# Ultrathin 2 nm gold as impedance-matched absorber for infrared light

Niklas Luhmann 🄳 [1,3], Dennis Høj[2,3], Markus Piller[1], Hendrik Kähler[1], Miao-Hsuan Chien[1], Robert G. West[1], Ulrik Lund Andersen 🄳 [2] & Silvan Schmid 🄳 [1✉]

Thermal detectors are a cornerstone of infrared and terahertz technology due to their broad spectral range. These detectors call for efficient absorbers with a broad spectral response and minimal thermal mass. A common approach is based on impedance-matching the sheet resistance of a thin metallic film to half the free-space impedance. Thereby, one can achieve a wavelength-independent absorptivity of up to 50%. However, existing absorber films typically require a thickness of the order of tens of nanometers, which can significantly deteriorate the response of a thermal transducer. Here, we present the application of ultrathin gold (2 nm) on top of a surfactant layer of oxidized copper as an effective infrared absorber. An almost wavelength-independent and long-time stable absorptivity of 47(3)%, ranging from 2 µm to 20 µm, can be obtained. The presented absorber allows for a significant improvement of infrared/terahertz technologies in general and thermal detectors in particular.

[1] Institute of Sensor and Actuator Systems, TU Wien, Gußhausstraße 27-29, 1040 Vienna, Austria. [2] Department of Physics, Technical University of Denmark, Fysikvej, 2800 Kongens Lyngby, Denmark. [3] These authors contributed equally: Niklas Luhmann, Dennis Høj. ✉email: silvan.schmid@tuwien.ac.at

Efficient absorbers of electromagnetic radiation are a fundamental element of infrared (IR)/terahertz (THz) technology, and of specific importance for the development of sensitive detectors. In particular, thermal IR/THz detectors have remained the only technology covering the entire spectral range from the visible to the far-IR (THz) regime, facilitating applications ranging from spectrochemical analysis to security and astronomy[1–6]. Thereby, they exploit the conversion of the absorbed photo-thermal power to either a change in electrical resistance or electric potential, as in bolometers or pyroelectrics and thermocouples, respectively[1]. More recently, microelectromechanical and nanoelectromechanical systems (MEMS/NEMS) have demonstrated exceptional potential as IR and THz detectors due to strong photothermally-induced detuning of their mechanical resonance frequency[7–14].

Despite such variations in detection paradigms, the development of efficient absorbers remains a crucial task in the development of high-performance IR/THz detectors. A suited absorber, in this case, should provide long-term stability, a broad and flat spectral response, while having an negligible thermal mass. This has been the focus for advancement in IR/THz detection for decades and has led to a reduction in effective thickness of the absorber to the order of 10 nm in the present day[1].

At this scale, contemporary solutions are numerous: from antenna structures[11,15–17] to metamaterials[18–20], which promise absorptivities up to 100% but are always limited by their resonance bandwidth. A progressive solution to improve the spectral range of detection has been to use a stack of plasmonic structures with differing lateral size; nonetheless, the bandwidth of these sensors remains limited e.g., from 0.8 to 1.3 THz[21]. A most recent thrust toward the ultimate limit of uncooled detection has been to employ the exceptional properties of 2D materials as graphene[22], even making the detector itself the primary absorber, for example, as uncooled NEMS resonator[8]. However, due to graphene's low absorptivity in the near-IR to mid-IR of only 2.3%[23,24], modifications using plasmonic metastructures are still required[25], limiting the absorber again to a certain bandwidth.

In the modern age of nano-scale and atomic-scale detectors, we may need to return to a classic, old-fashioned approach: to engineer the sheet resistance of a thin metal such that it matches half the free space impedance $\sqrt{\mu_0/4\varepsilon_0} \approx 188\,\Omega$[26–28]. Based on the theory introduced by Woltersdorff[28] and further extended by Hadley et al.[29] and Hilsum[30], a wavelength-independent absorptivity of up to 50% can be achieved, assuming the optical constants $n$, $k$ are approximately equal — which, for metals such as gold, is only valid in the far-IR.

Many approaches using thin layers of e.g., bismuth, silver (Ag), or platinum[31–36], unseeded metastructures[37], and alloys, such as titanium nitride (TiN), nickel chromium (NiCr)[9,10], and indium tin oxide (ITO)[38], have been successfully tested for this purpose. However, some alloys/metals are prone to oxidization, which can affect the absorptivity over time. Regarding the thickness needed to match the desired impedance, alloys such as TiN, with an optimum of 14 nm, are often in the same dimension as the detecting element itself[7,9,10]. Other metals, such as gold, need a comparably large thickness to reach the percolation threshold, setting a lower limit for thin-film thickness at the insulator-to-metal transition, which normally makes them too conductive to match the necessary sheet resistance[39]. Alternatively, an impedance-matched absorber based on chemically doped graphene has been demonstrated in the THz regime[40]. However, the response towards shorter wavelengths has not been investigated. While graphene is ultimately the thinnest material, the fabrication and integration of such an absorber is relatively complex.

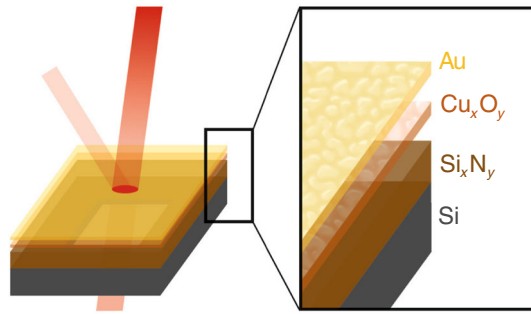

**Fig. 1 Illustration of the fabricated layers and probing direction.** The samples are based on 50 nm $Si_xN_y$ membranes comprising an initially 1.2 (2) nm sputtered and further naturally oxidized copper layer. For characterization of optimal thickness, varying Au layers were deposited on top using physical vapor deposition.

Recently, it has been shown that it is possible to fabricate ultrathin metal films (UTMF) of Ag[41] and Au[42] far below the typical percolation limit, using a surfactant layer of oxidized copper.

Here, we demonstrate the application of this technique to fabricate an ultrathin Au layer as an efficient, broad spectral impedance-matched IR absorber. Figure 1 shows a schematic illustration of the fabricated samples used for this study. Due to the differing optical and electrical properties of ultrathin gold compared to bulk values, it is possible to gain an almost wavelength-independent absorptivity of 47(3)% down to below 2 μm with films only 2 nm thick. With its low thermal mass, good stability, and easy producibility, the presented absorber allows for a significant improvement of IR/THz technologies in general and thermal detectors in particular.

## Results

**Theoretical model of impedance-matched absorption.** The model of impedance-matched absorption[28–30] is based on the assumption that the refractive index $n$ and extinction coefficient $\kappa$ of the metal are to be equal. From our knowledge the validity of this criteria has yet not been discussed, especially in regard to the limiting wavelength where it can be applied. To get a better understanding of when this assumption is valid, the Drude model is used and rewritten in terms of plasma frequency $\omega_p$ and electrical resistivity $\rho$:

$$\hat{\varepsilon} = \varepsilon_1 + i\varepsilon_2 = 1 - \frac{1}{\omega^2/\omega_p^2 + i\varepsilon_0\rho\,\omega}, \qquad (1)$$

where $\omega$ is the angular frequency of the optical field. The refractive index and extinction coefficient is then given by $n = \sqrt{|\hat{\varepsilon}| + \varepsilon_1}$ and $\kappa = \sqrt{|\hat{\varepsilon}| - \varepsilon_1}$, respectively. It can clearly be seen, in order for the assumption ($n = \kappa$) to be valid, the imaginary part of the relative permittivity $\varepsilon_2$ must dominate. This is true when

$$\omega \ll \varepsilon_0\,\rho\,\omega_p^2. \qquad (2)$$

In this regime, the Drude model can then be simplified to

$$\hat{\varepsilon} \approx \frac{1}{\varepsilon_0^2\rho^2\omega_p^2} + i\frac{1}{\varepsilon_0\rho\omega}, \qquad (3)$$

where it can be seen that the real part is limited to some finite value; whereas, the imaginary part is increasing for longer wavelengths. For gold, assuming bulk values $\rho = 2.2 \times 10^{-8}\,\Omega\cdot$m and $\omega_p = 2\pi\cdot2.1$ PHz, this limit is approximately at a wavelength of 56 μm. Below this limit it should not be possible to achieve high absorptivity, unless the material parameters change, which is

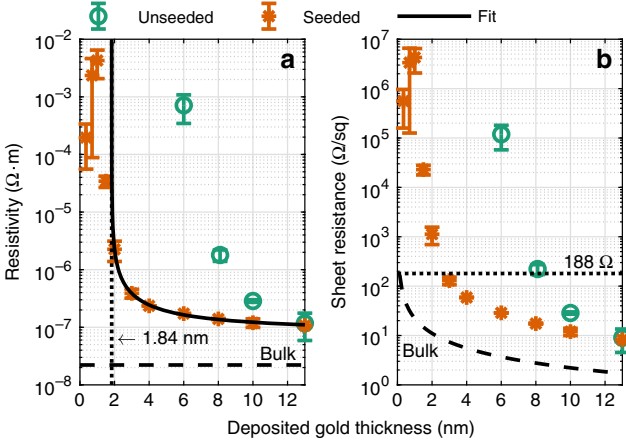

**Fig. 2 Electrical properties of UTMFs. a** Measured resistivity of seeded and unseeded Au layers as a function of deposited thickness. Due to the oxidized copper seed layer a metallic behavior of Au can be obtained down to 2 nm. The resistivity can be well described by the scattering hypothesis Eq. (4) and is strongly governed by grain-boundary and surface scattering $\propto d^{-1}$. In order to fit the data, a 1.84 nm offset is included to compensate the percolation threshold and uncertainty of effective thickness. **b** Resulting sheet resistance of the same samples. Thus, the optimal thickness for impedance-match to 188 Ω is expected around 2.5 nm. Bulk gold data taken from[60]. Error bars represent the standard deviation obtained from ten measurements on each sample. Source data are provided as a Source Data file.

indeed the case for UTMFs[39]. Especially, regarding electrical properties, the resistivity can be many factors of magnitude higher than compared to bulk[43–45]. To a certain extent, this can be described by the so-called scattering hypothesis; whereby, the materials' resistivity is defined as a sum of scattering contributions[46]

$$\rho = \rho_0 + \rho_{GB} + \rho_{SS} + \rho_{SR}, \qquad (4)$$

where $\rho_0$ is the bulk resistivity, $\rho_{GB} \propto D^{-1}$ is the grain-boundary contribution, $\rho_{SS} \propto d^{-1}$ is the surface scattering contribution, and $\rho_{SR} \propto d^{-3}$ is the roughness contribution. Here, $d$ is the metal thickness and $D$ the mean grain width of the metal film. For thin films, one can approximate $D$ to be equal to $d$[45]. However, with increasing thickness, a limiting grain size of $D_\infty$ is reached. The grain-boundary contribution can therefore be extended to $\rho_{GB} \propto 1/D_\infty + 1/(Cd)$, where $D_\infty$ is often found to be limited up to ~20 nm, respectively the range of the materials' electron mean free path. The dimensionless factor $C$ typically ranges from 0.5 to 1[45]. Below percolation, ohmic bridges and tunneling effects govern the resistivity, which can be described by e.g., Monte-Carlo simulations or the filamentary vibron quantum percolation model[47,48], but however, are beyond the scope of this study. Collectively, all these resistivity contributions lead to broaden the absorptivity bandwidth of thin films down to shorter wavelengths, as described by Eq. (2).

**Resistivity and sheet resistance of UTMF.** Figure 2 shows the measured resistivity and corresponding sheet resistance of seeded vs. unseeded gold as a function of the deposited layer thickness. Consistent with previous studies on ultrathin copper and gold films[41,43,49], the resistivity can be partly fitted by the scattering hypothesis Eq. (4) including a variable offset to compensate the percolation threshold and potential uncertainty of the effective film thickness. The data can be well described by the model down to 2 nm, including a positive offset of 1.84 nm and is governed by the grain-boundary and surface scattering term $\propto d^{-1}$. The offset

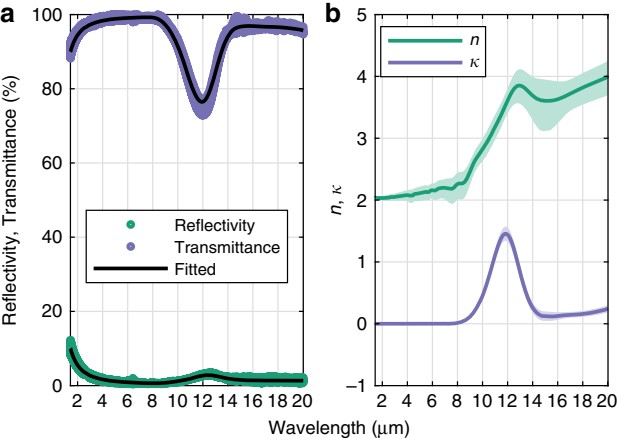

**Fig. 3 Optical properties of support layer. a** Average of ten individual FTIR measurements on 50 nm silicon nitride membranes. **b** Fitted optical constants using a general matrix model[55] by grouping measured data into fixed wavelength points, from which the optical constants were estimated individually (see Supplementary Methods). The shaded region indicates the uncertainty estimated as a 50% increase in sum of squared residuals. Source data are provided as a Source Data file.

can be related to the percolation threshold. It is exactly between 1.5 and 2 nm where the resistivity drops below the insulator-to-metal transition value of around 6 μΩ·m, defined by Ioffe-Regel[50]. Thus, percolation must occur in between those samples. As can be seen, Au layers below 1.5 nm show an unexpected reverse trend of the resistivity with increasing thickness. This effect has previously been observed for UTMFs below percolation and can be related to the growth of islands and isolated atoms, which act as additional scattering centers[51,52] (For more details see Supplementary Note 3).

Compared to the bulk resistivity of Au with $2.2 \times 10^{-8}$ Ω·m, the 13 nm thick layer converges to a three times higher value. As mentioned, the grain-boundary scattering term includes a limiting grain size factor of $1/D_\infty$, which significantly lifts the resistivity up to a thickness in the range of the electron mean free path ~20 nm. Thus, with respect to previous studies[45], such an increased value can be expected. Regarding the measured sheet resistance, the optimal thickness for matching with half the free space impedance to 188 Ω is expected around 2.5 nm Au.

To investigate whether there is a potential contribution of the oxidized copper used as seed layer, additional experiments with varying Cu thicknesses revealed that bare Cu could be also used as an impedance-matched absorber but only if measured directly after deposition. However, due to oxidization over time[41,53], all reference samples with oxidized copper showed a significant increase of resistivity back to the insulating state, and consequential loss of absorptivity. Regarding the presented data, all samples used were taken from a single processed wafer with an initial deposited Cu thickness of 1.2(2) nm. A comparison between seeded and unseeded membranes of this batch showed no significant dependence of the electrical or optical properties on the oxidized seed layer thickness.

**Plasma frequency and wavelength-independent absorptivity.** In order to obtain a proper fit for the transmittance and reflectivity of ultrathin Au, optical data for the LPCVD $Si_xN_y$ was needed. This was extracted from multiple Fourier transformed IR (FTIR) spectroscopy measurements on bare LPCVD $Si_xN_y$ membranes shown in Fig. 3a. Here, the drop of transmittance and slight increase of reflectivity between 9 to 12 μm is an intrinsic material behavior of $Si_xN_y$[54]. Using an optical model[55] adapted to a single

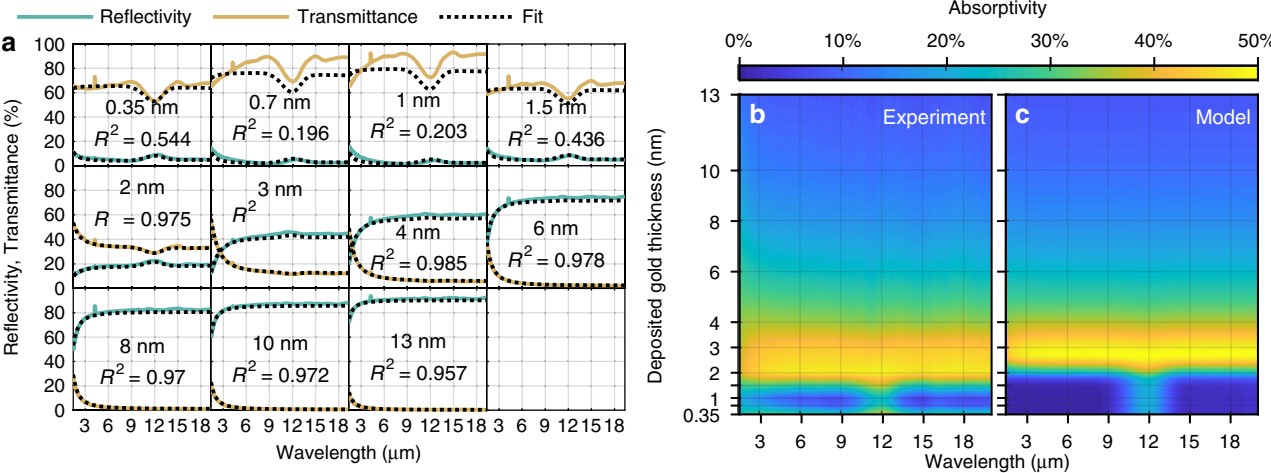

**Fig. 4 Optical Properties of the UTMFs. a** Measured transmittance and reflectivity of (seeded) Au layers, obtained by FTIR spectroscopy. All spectra are fitted by the given Drude Model Eq. (1), including the measured resistivity. $R^2$ is the coefficient of determination for each fit. As for thicker layers, all spectra are in good agreement with the optical properties of a metallic film, the layers below 2 nm show a divergence, due to the insulator-to-metal transition. For the metallic-like layers, the obtained plasma frequency remains constant with $\omega_p \sim 2\pi \cdot 3.0(7)$ PHz. **b** Absorptivity as a function of deposited gold thickness and wavelength obtained experimentally from FTIR measurements. The data has been linearly interpolated for the plot. Each horizontal grid line corresponds to a sample. All measured layers were evaporated on 50 nm $Si_xN_y$ membranes comprising an oxidized copper seed layer. **c** Calculated absorptivity using the fitted Drude parameters and measured resistivity via Eq. (1) and extracted optical properties of $Si_xN_y$. Source data are provided as a Source Data file.

$Si_xN_y$ layer, to predict the transmittance and reflectivity, a non-linear fit was done at each wavelength separately, in order to estimate the optical constants (see Supplementary Methods for more details). To reduce the uncertainty of the fit, the measured data were grouped into spectral blocks with 50% overlap. This increases the amount of data per spectral point at the cost of spectral resolution. The result is shown in Fig. 3b with 618 spectral points. Note that the fit is not Kramers-Kronig constrained[56], which was not possible due to the limited spectral range of the measurements. However, for the purpose of this paper the data is deemed acceptable.

Figure 4a shows the measured transmittance and reflectivity of all seeded gold layers. The data is fitted by using the above-mentioned optical model, in combination with the Drude model from Eq. (1), and including the previous extracted optical properties for $Si_xN_y$. As can be seen from the $R^2$ coefficient of determination, the thicker layers show a metallic behavior and can be well fitted down to 2 nm, consistent with the percolation threshold obtained before. For those thicknesses, the extracted plasma frequency with $\omega_p \approx 2\pi \cdot 3.0(7)$ PHz is slightly increased compared to bulk but remains constant within its uncertainty. Due to the loss of metallic behavior, the model can not be effectively applied below percolation (2 nm), which can be clearly seen by the dropping $R^2$ coefficient. In this region, an increase of the transmittance from the 0.35 nm to the 0.7 and 1 nm sample can be observed. Previous studies on thin Au layers around the percolation threshold have confirmed this antireflection phenomena[57], whose origin lies in the divergence of the dielectric constant $\varepsilon_1$ in that region[39]. One could, therefore, suggest the use of ultrathin Au below percolation as potential anti-reflection coating.

Figure 4b shows the measured, linearly interpolated absorptivity (1—reflectivity—transmittance), plotted over the wavelength and deposited Au thickness. One can clearly determine a maximum absorptivity of 40–50% between the 2 nm and 4 nm Au layer, with a slight decrease towards smaller wavelengths. The absorption peak with a maximum at 12 μm below a gold thickness of 2 nm is a contribution from the supporting $Si_xN_y$

substrate, as can be seen in Fig. 3. The slight increase in absorptivity for the thinnest Au layer of 0.35 nm could be related to the observed reverse trend of the resistivity as can be seen in Fig. 2. A comparison of the average transmittance and reflectivity, as function of the sheet resistance, to the general impedance-match theory[28–30] can be found in the Supplementary Note 1. Regarding long-term stability, subsequent measurements verified, that the 2 nm samples showed no significant change in absorptivity over a period of five month being stored under ambient conditions. Investigations of similar Au thin films on oxidized Cu also showed a high stability[58].

Figure 4c presents the corresponding calculated absorptivity based on the previously obtained optical properties of bare $Si_xN_y$ and the relative permittivity (see Eq. (1)), comprising the measured resistivity and extracted plasma frequency. A comparison to the data shows a good agreement with a minimal offset of ~ 0.35 nm towards the optimal thickness, which lies in the uncertainty of the quartz sensor. With respect to the derived criteria Eq. (2), the increased resistivity and plasma frequency lead to an almost wavelength-independent high absorptivity, ranging from 2 μm up to the detection limit of 20 μm. It is expected that the given absorptivity remains constant also in the far-IR (THz) regime.

**Dominance of the imaginary part of relative permittivity.** In a final step, the dielectric functions of each metallic layer is extracted via Eq. (1) and plotted in Fig. 5. As can be clearly seen by the ratio $|\varepsilon_2/\varepsilon_1|$ (black line), for the 2 nm Au film the magnitude of the imaginary part $\varepsilon_2$ is larger than that of the the real part $\varepsilon_1$ over the entire spectral range, in good agreement with the derived criteria Eq. (2). Consequentially, this determines the lower spectral limit of the fabricated impedance-matched absorber to 2 μm. For thicker gold films, the $|\varepsilon_2/\varepsilon_1| > 1$ is reached with increasing wavelengths. Regarding the criteria itself, it should be possible to further broaden the wavelength-independent absorptivity by using a material with a larger plasma frequency and higher resistivity such as e.g., aluminum, or by other engineered

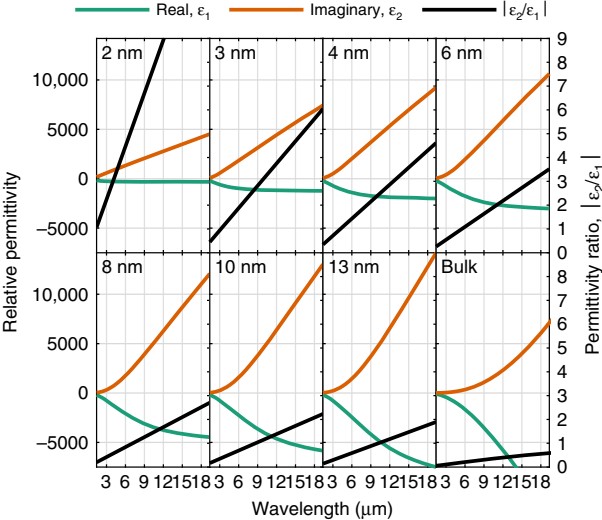

**Fig. 5 Relative permittivity of UTMFs.** Extracted relative permittivity for increasing gold layer thickness, based on the fitted Drude model Eq. (1). The solid black line represents the ratio of the imaginary vs real part of the permittivity. For thinner layers, where impedance-matched absorption occurs, the imaginary part strongly dominates with increasing wavelength, in good agreement with the derived criteria Eq. (2). The bulk data is for evaporated gold taken from the ref. [61]. Source code available on request.

materials such as doped semiconductors, which needs to be further investigated.

## Discussion

We have demonstrated the application of ultrathin 2 nm Au as a highly efficient, broad spectral, impedance-matched absorber. In good agreement with theoretical assumptions, it was possible to gain 47(3)% absorptivity over the entire near- and mid-IR range from 2 to 20 μm. According to the theory, this wavelength-independent absorptivity is expected to be also valid in the far-IR (THz) domain, as demonstrated by previous studies[40]. Electrical and optical analysis of the deposited Au layers demonstrated the significantly increased resistivity and impact on the optical properties of such ultrathin metal films, which broaden the lower limit of impedance-matched absorption to 2 μm. In this context, it was possible to approximately match the optimal sheet resistance of 188 Ω. The extracted dielectric functions verified, that for those samples the imaginary part of the relative permittivity is strongly enhanced compared to the real part, which in this region can be almost neglected. In the course of this study the optical constants of LPCVD $Si_xN_y$ in the range of 2 to 20 μm were obtained experimentally and further used to estimate the absorptivity. Comprising a small offset of the layer thickness, the calculated absorptivity is in well agreement with the experimental results. Furthermore, the obtained transmittance and reflectance below percolation < 1.84 nm indicate the potential use of ultrathin Au to be used as anti-reflection coating. Overall, considering its negligible small thermal mass, and constant broad spectral absorptivity, we suggest that the fabricated absorber is well-suited to be used in thermal IR and THz detector applications.

## Methods

**Fabrication of ultrathin gold**. Figure 1 shows a schematic illustration of the fabricated samples used for this study. All experiments were conducted on 50 nm thin silicon nitride ($Si_xN_y$) deposited by low-pressure chemical vapor deposition (LPCVD), acting as a support layer with minimal absorptivity. The 2.5 mm × 2.5 mm squared membranes were structured by standard UV lithography and backside released in potassium hydroxide (KOH). For the ultrathin gold fabrication, a Cu

layer was deposited by sputter deposition using a Von Ardenne LS 730S. The deposition rate was set to 1.5 Å s$^{-1}$, extrapolated from several test depositions. For all samples used in this work, an initial Cu thickness of 1.2(2) nm was deposited. To ensure a smooth and clean surface, the $Si_xN_y$ membranes were plasma cleaned using Argon-based reactive ion etching in the same vacuum chamber, immediately before Cu deposition. Following Maniyara et al.[42], the samples were then stored in air for one day to undergo oxidation. During this process, oxidization of Cu can lead to an increase of volume up to 68% (assuming formation of $Cu_2O$[42]). Thus, the resulting seed layer thickness is expected to be slightly larger than the initial deposited film. In a final step, gold was evaporated from a tungsten boat with a comparably low rate of 0.3 Å s$^{-1}$ at $3 \times 10^{-8}$ mbar. The deposition rate and extracted nominal thickness was monitored by a quartz resonating sensor. The thin-film morphology was analysed by means of atomic force microscope images of the 2 nm absorber and bare $Si_xN_y$, showing a smooth and uniform surface without confined grains (see Supplementary Note 2). Thus, the fabricated UTMFs above the extracted percolation threshold of 1.84 nm can be treated as a continuous metal film.

**Optical characterization**. All optical spectra were recorded via FTIR spectroscopy conducted with a Bruker Tensor 27. In order to minimize systematic deviations, transmittance and reflectivity measurements were performed within one measurement using a specified A 510/Q-T set-up and an aperture of 2 mm. In order to model the optical behavior, a general matrix method to predict the transmittance and reflectivity has been implemented[55] (see Supplementary Methods).

**Electrical characterization**. The resistivity and sheet resistance were obtained by a homemade four-point-probe setup made of a cylindrical probe-head provided by Jandel, a Keithley 6221 current source, and a Keithley 2182 A nanovoltmeter. The probe-head was attached to a load bending beam Burster 8511–5050 to monitor the contact force applied on the surface during measurements. All samples were measured in a current range of $10^{-7}$–$10^{-3}$ A and maximum contact force of 2.2 N.

## Data availability

The source data underlying the presented Figs. 2–5 including Supplementary Figs. 1, 3 are provided as a Source Data file and available on an online repository[59].

## Code availability

All custom codes used for data processing underlying Figs. 2–5 and Supplementary Fig. 1 are available from the authors upon request.

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

## Acknowledgements
We wish to acknowledge the support of Philipp Altpeter and Philipp Paulitschke for giving access to the clean room facilities of LMU Munich. Furtherstill, we want to thank our technicians Sophia Ewert, Michael Buchholz, and Patrik Meyer for their ongoing support. Special thanks to Christoph Eisenmenger-Sittner from the physical department for condensed matter at TU Wien for providing access to their homemade four-point-probe station. And we would like to thank Michael Feiginov and Pedram Sadeghi for the discussions. In addition we would like to thank Jonas Hafner for his support on the AFM. D.H. and U.L.A. acknowledge support from the Villum foundation and the Danish National Research Foundation. This work has received funding from the European Research Council under the European Union's Horizon 2020 research and innovation program (Grant Agreement-716087-PLASMECS).

## Author contributions
Fabrication and measurements were performed by N.L., while modeling and data analysis was done by D.H. and N.L; results and data interpretation were discussed by N.L., D. H., M.P., H.K., M.-H.C., R.G.W., and S.S. Further, N.L., D.H., M.P., H.K., M.-H.C., R.G. W., S.S., and U.L.A. have contributed to writing of the manuscript. U.L.A. and S.S. supervised the research.

## Competing interests
The authors declare no competing interests.
