## [Peer Review File · Nature Communications]

Reviewers' comments:

Reviewer #1 (Remarks to the Author):

This is a quite straightforward paper: the authors implement a 2 nm thick gold layer whose surface impedance is close to impedance matched to free-space in the mid-infrared range, enabling broadband absorptivity. The results make sense, and the main contribution of the authors is the realization of this uniform layer with small thickness. I am not convinced that this contribution belongs to Nature Communications, as the physics involved is rather straightforward and there is nothing particularly surprising in the overall response. The authors may want to demonstrate a Salisbury configuration to show a more narrowband absorber with 100% absorptivity. Anyhow, I believe that the paper may belong to a more specialized journal.

Reviewer #2 (Remarks to the Author):

The paper describes an interesting approach to achieve broadband absorption in the infrared, to be applied to relevant devices, such as photodetectors. Results are novel, including the demonstration of the functionality of ultrathin gold films (fabricated using a previously published technique). I have several remarks, which should be addressed before publication:

- The authors deposit metallic copper as surfactant (seed layer). The copper then oxidizes as the surfactant coated substrate is left in open atmosphere before gold deposition. My suggestion is not to call the surfactant (seed) layer "copper oxide" or "Cu_xO_y", but more appropriately call it "oxidized copper" through the whole paper. Also because the initial copper layer may be only partly and not fully oxidized.

- In the abstract and paper text, they also mention that the surfactant layer is 1.2 nm. This is the initial copper thickness and not the final oxidized copper under the gold layer. As mentioned before, copper will oxidize and increase its volume accordingly, thus the surfactant layer is very likely thicker. The increase in volume (Thickness) depends on the type of oxide/s formed (e.g. CuO, Cu₂O, etc...) and the level of oxidation (ratio between forming oxides and metal). I would suggest stating in the paper that 1.2nm is the initial metal thickness and that the oxidized copper surfactant (seed) layer before gold deposition is likely to be larger because of the volume increase due to oxidation. I would also remove the thickness value, i.e. 1.2 nm, in the abstract.

- The sentence "The deposition rate was set to 1.5 °A s⁻¹, extrapolated from several test depositions, resulting in a seed layer thickness of 1.2(2) nm". What do they mean with "(2)" after "1.2"?

- The sentence "Regarding the presented data, we found no significant influence on the electrical or optical properties by the seed layer used." What do exactly the authors mean with this sentence? That the gold film properties do not depend significantly on the seed layer? If so, do they mean that they do not depend on the thickness of seed layer? Up to what initial copper thickness? 1.2 nm?

- The sentence "Other metals, such as gold, demonstrate strong percolation effects,....." is not clear to me. Percolation is when the film somehow shows a conductive behaviour. Gold without surfactant (seed) layer does not percolate at small thicknesses, as it is also shown in the paper. So I am not sure why the authors say that gold demonstrates strong percolation effects. Maybe I am missing something.

- The 2 nm gold layer: what is its morphology? It behaves as a "continuous" metallic layer but it may not actually be. A percolated and morphologically not uniform metallic film can behave optically as a continuous film with different refractive index constants when the structural features are

subwavelength. If the authors did not measure the morphology of the 2nm gold film, they may want to specify this in the paper "The fact that the 2 nm Au layer behaves as if it were continuous does not necessarily mean that it is uniform morphologically.....as a subwavelength percolated metallic film behave as a uniform film with different optical constants.....effective medium approximation"

- Is 50% absorption enough for many applications? Also, referring to this, when they talk about the use of resonant plasmonic structures they may want to specify what absorption one can obtain in that case and over which wavelength range.

- Fig. 2: it seems they have a lower resistivity and lower sheet resistance for 0.5 nm than 1 nm gold films. Is this correct? If it is, do the authors have an explanation for this? Also, considering that this does not seem not reflected in the optical behavior observed.

Reviewer #3 (Remarks to the Author):

Review:

This is beautiful and significant work on the absorption of thin films in the far and near IR at almost perfect impedance matching. Congratulations to the authors on a job well done measuring this. It should be published for sure and is very significant result.

There are some major issues with the manuscript presentation that need to be fixed prior to publication.

First, the authors are confusing two things: 1) Bolometric response (change in resistance with change in temperature due to heating by IR) 2) Absorption coefficient. The absorption coefficient is the MAIN result of this paper. It is near perfect (well, 50%) for a thin film as they have shown. However, they are trying to "sell" this result as a bolometric detector. This is not necessary in my opinion. Even #2 alone is a major significant result of general and broad appeal. And for example, if the absorption is perfect but there is no change in resistance with absorbed power, or little change, then #1 is not important technologically for detectors. On the other hand, if the bolometric response is strong (i.e. there is a large change in resistance with absorbed power), then even if the absorption coefficient is not perfect, the device can still be a good detector. Does Au film have any dR/dT at all or is it just planned to be used as an absorber and the thermometer another element? If so the thermal mass of the element has to be taken into account, not just the thin film (see last sentence of paper re thermal mass). So, the authors need to split these two results out clearly, and not confuse #1 with #2.

Second, the authors seem to have missed in the literature how this has already been done over a broad range in the THz band in 2 papers (Brown, E. R., Zhang, W. D., Chen, H., & Mearini, G. T. (2015). THz behavior of indium-tin-oxide films on p-Si substrates. *Applied Physics Letters*, 107(9), 091102. <https://doi.org/10.1063/1.4929755> using ITO and Pham et al Pham, P. H. Q., Zhang, W., Quach, N. V., Li, J., Zhou, W., Scarmardo, D., ... Burke, P. J. (2017). Broadband impedance match to two-dimensional materials in the terahertz domain. *Nature Communications*, 8(1), 2233. <https://doi.org/10.1038/s41467-017-02336-z> using graphene). Furthermore, the authors did not plot their absorption coefficient (and reflection and transmission coefficient) vs. sheet resistance from their measured data, and compared to theory, which they definitely should do, together with theory, see for example fig. 2,4 in Pham et al. See Zanotto, S., Bianco, F., Miseikis, V., Convertino, D., Coletti, C., & Tredicucci, A. (2017). Coherent absorption of light by graphene and other optically conducting surfaces in realistic on-substrate configurations. *APL Photonics*, 2(1), 016101 for the theory part.)

Then, they need to say how their work is different than those 2 references. I think it is because this work is in the IR and the 2 refs. are THz, but the authors need to make this conclusion, not me as reviewer. Especially since there is a lot of talk about THz in this paper.

A minor 3rd point is, what is the dip due to in the figure 3?

So in conclusion , great experimental work, high significance.
Needs to address #1,2 prior to publication.

Reviewer #1 Remarks and Answers

“This is a quite straightforward paper: the authors implement a 2 nm thick gold layer whose surface impedance is close to impedance matched to free-space in the mid-infrared range, enabling broadband absorptivity. The results make sense, and the main contribution of the authors is the realization of this uniform layer with small thickness. I am not convinced that this contribution belongs to Nature Communications, as the physics involved is rather straightforward and there is nothing particularly surprising in the overall response.”

Thank you for your feedback. We agree that the physics behind such an impedance-matched absorber is rather straightforward; however, its limit with respect to the wavelength has from our knowledge not been studied. Furthermore, impedance-matched absorbers can either be fabricated with metal thin films that are comparably thick, or by 2D materials, such as graphene, but that are hard to produce on a larger scale. The novelty we present in our paper is an impedance-matched absorber based on an ultimately thin metal film made of gold, which is easy to fabricate on wafer-scale, with an ideal flat response starting from 2 μ m, reaching the theoretical limit of 50% absorptivity, which by theory is expected to be also valid in the THz domain. In particular, the mid-IR regime hosts a wealth of interesting applications ranging from

spectrochemical analysis to thermal imaging. We therefore believe that our results are of interest to a wide and multidisciplinary audience that can be reached by Nature Communications.

“The authors may want to demonstrate a Salisbury configuration to show a more narrowband absorber with 100% absorptivity.”

We thank the reviewer for the suggestion. Indeed, our absorber could readily be implemented in a Salisbury configuration. We specifically designed a general-purpose absorber with a flat and broad spectral response. Our absorber is directly of interest for applications e.g. for infrared spectroscopy that require a flat and broad spectral range. And it can also be implemented in other designs, including a Salisbury configuration, if an improved absorption is desired over a defined spectral range. However, additional configurations can only increase the efficiency by a maximum factor of two.

Reviewer #2 Remarks and Answers

“The paper describes an interesting approach to achieve broadband absorption in the infrared, to be applied to relevant devices, such as photodetectors. Results are novel, including the demonstration of the functionality of ultrathin gold films (fabricated using a previously published technique). I have several remarks, which should be addressed before publication:

Thank you for your positive feedback.

The authors deposit metallic copper as surfactant (seed layer). The copper then oxidizes as the surfactant coated substrate is left in open atmosphere before gold deposition. My suggestion is not to call the surfactant (seed) layer “copper oxide” or “Cu_xO_y”, but more appropriately call it “oxidized copper” through the whole paper. Also because the initial copper layer may be only partly and not fully oxidized.”

We acknowledge this thoughtful input and will follow the suggestion throughout the manuscript.

“In the abstract and paper text, they also mention that the surfactant layer is 1.2 nm. This is the initial copper thickness and not the final oxidized copper under the gold layer. As mentioned before, copper will oxidize and increase its volume accordingly, thus the surfactant layer is very likely thicker. [...] I would suggest stating in the paper that 1.2nm is the initial metal thickness and that the oxidized copper surfactant (seed) layer before gold deposition is likely to be larger because of the volume increase due to oxidation. I would also remove the thickness value, i.e. 1.2 nm, in the abstract.”

This is a valid point and we have made the appropriate changes in the abstract and the method section in the manuscript.

“The sentence “The deposition rate was set to 1.5 \AA s^{-1} , extrapolated from several test depositions, resulting in a seed layer thickness of 1.2(2) nm”. What do they mean with “(2)” after “1.2”?”

We use parentheses as our preferred notation of the uncertainty of a value based on its least significant digit. In this example 1.2(2) nm could also be written as 1.2 ± 0.2 .

“The sentence “Regarding the presented data, we found no significant influence on the electrical or optical properties by the seed layer used.” What do exactly the authors mean with this sentence? That the gold film properties do not depend significantly on the seed layer? If so, do they mean that they do not depend on the thickness of seed layer? Up to what initial copper thickness? 1.2 nm?”

We thank the reviewer for pointing out this lack of clarity. Yes, what we meant to say is that the gold film properties do not depend on the seed layer thickness. As mentioned in the manuscript, for the optimization of the seed layer, different thicknesses have been tested. In this context, even an initial layer of 0.5 nm Cu can sufficiently tune the percolation threshold of gold. However, due to the very short deposition time and the fact that the used sputter machine does not allow for the stage to be rotated during deposition, there is a slight gradient of the thickness across the wafer being processed. Therefore, for practical reasons we choose to set the initial seed layer thickness to a more reproducible value of 1.2nm.

“The sentence “Other metals, such as gold, demonstrate strong percolation effects,.....” is not clear to me. Percolation is when the film somehow shows a conductive behavior. Gold without surfactant (seed) layer does not percolate at small thicknesses, as it is also shown in the paper. So I am not sure why the authors say that gold demonstrates strong percolation effects. Maybe I am missing something.”

Thank you for pointing this out. Yes, we meant to explain that metals, such as gold, need a comparably large thickness to reach the percolation when compared to other metals such as chromium. We have revised the manuscript in the introduction to clarify this point.

“The 2 nm gold layer: what is its morphology? It behaves as a “continuous” metallic layer but it may not actually be. A percolated and morphologically not uniform metallic film can behave optically as a continuous film with different refractive index constants when the structural features are subwavelength. If the authors did not measure the morphology of the 2nm gold film, they may want to specify this in the paper “The fact that the 2 nm Au layer behaves as if it were continuous does not necessarily mean that it is uniform morphologically.....as a subwavelength percolated metallic film behave as a uniform film with different optical constants.....effective medium approximation”

Thank you for raising this point. In the framework of this study, the morphology has been measured with an AFM but not added to the manuscript. Following your argument, we see the necessity of explaining the morphology and therefore made appropriate comments to this regard in the manuscript under the method section and added a new section to the Supplementary Information. Therefore, new AFM scans with higher resolution were made for this revision, which clearly show, that the UTMF surface looks generally smooth and without any confined grains. The AFM analysis showed a slight decrease in roughness for the UTMF surface as compared to the bare SiN surface, which supports the conclusion of the UTMF as a continuous and uniform layer.

Another evidence that the UTMF represents a continuous and uniform film comes from the theoretical modelling. Initially, we used an effective medium approximation to fit the data. However, in the presented work we finally gained a much better match to the measured data when assuming the UTMF to be an uniform “Drude-like” metal.

“Is 50% absorption enough for many applications? Also, referring to this, when they talk about the use of resonant plasmonic structures they may want to specify what absorption one can obtain in that case and over which wavelength range.”

Thank you for the question. For many applications, such as IR spectroscopy, a broad and flat response is more important than an improvement of the absorption by a factor of 2 (from 50% up to 100%). Thus, from our point of view, reverting to a classical approach and providing a broad and flat response over the entire IR spectrum, although being limited to 50% efficiency, seemed to be more advantageous. As discussed by reviewer #1, it is always possible to improve the absorption up to 100% by implementing our absorber e.g. in a Salisbury configuration. We have heeded the reviewer’s suggestion to specify this point in the introduction in order to underline the benefit of the fabricated UTMF absorber.

“Fig. 2: it seems they have a lower resistivity and lower sheet resistance for 0.5 nm than 1 nm gold films. Is this correct? If it is, do the authors have an explanation for this? Also, considering that this does not seem not reflected in the optical behavior observed.”

Indeed, some samples showed an unexpected decrease of the resistivity below percolation. With our 4-Point-Probe setup, it was comparably difficult to gain accurate data in the insulating region, which is also reflected by the large error-bars in the plot. The lower resistivity of the thinnest Au film is not systematic but rather an outlier.

Reviewer #3 Remarks and Answers

“This is beautiful and significant work on the absorption of thin films in the far and near IR at almost perfect impedance matching.”

Congratulations to the authors on a job well done measuring this. It should be published for sure and is very significant result.

Thank you for the positive feedback.

There are some major issues with the manuscript presentation that need to be fixed prior to publication.”

First, the authors are confusing two things: 1) Bolometric response (change in resistance with change in temperature due to heating by IR) 2) Absorption coefficient. The absorption coefficient is the MAIN result of this paper. It is near perfect (well, 50%) for a thin film as they have shown. However, they are trying to “sell” this result as a bolometric detector. This is not necessary in my opinion. Even #2 alone is a major significant result of general and broad appeal. And for example, if the absorption is perfect but there is no change in resistance with absorbed power, or little change, then #1 is not important technologically for detectors. On the other hand, if the bolometric response is strong (i.e. there is a large change in resistance with absorbed power), then even if the absorption coefficient is not perfect, the device can still be a good detector. Does Au film have any dR/dT at all or is it just planned to be used as an absorber and the thermometer another element? If so the thermal mass of the element has to be taken into account, not just the thin film (see last sentence of paper re thermal mass). So, the authors need to split these two results out clearly, and not confuse #1 with #2.”

Thank you for your response. We want to underline that the scope of our work was exclusively based on the development of a general-purpose ultrathin impedance-matched absorber, with minimal thermal mass. While one potential application for our absorber could be for IR/THz detectors, it is not limited to it. It's a good point, our absorber film might have a decent bolometric behavior. However, we have not investigated its behavior as a transducer (dR/dT). In this context, we specifically discussed the thermal mass of the fabricated absorber itself and did not include a discussion of the thermal mass of e.g. an attached transducer, as this strongly depend on the specific design of a potential application. We thank the reviewer to point out the necessity of clarifying this general point and made changes in the abstract and introduction.

“Second, the authors seem to have missed in the literature how this has already been done over a broad range in the THz band in 2 papers (Brown, E. R., Zhang, W. D., Chen, H., & Mearini, G. T. (2015). THz behavior of indium-tin-oxide films on p-Si substrates. Applied Physics Letters, 107(9), 091102. <https://doi.org/10.1063/1.4929755> using ITO and Pham et al Pham, P. H. Q., Zhang, W., Quach, N. V., Li, J., Zhou, W., Scarmardo, D., ... Burke, P. J. (2017). Broadband impedance match to two-dimensional materials in the terahertz domain. Nature Communications, 8(1), 2233. <https://doi.org/10.1038/s41467-017-02336-z> using graphene).”

We want to thank the reviewer for the recommend papers, which are indeed very important to be mentioned in this work. We are citing them now in the paper.

“Furthermore, the authors did not plot their absorption coefficient (and reflection and transmission coefficient) vs. sheet resistance from their measured data, and compared to theory, which they definitely should do, together with theory, see for example fig. 2,4 in Pham et al. See Zanotto, S., Bianco, F., Miseikis, V., Convertino, D., Coletti, C., & Tredicucci, A. (2017). Coherent absorption of light by graphene and other optically conducting surfaces in realistic on-substrate configurations. APL Photonics, 2(1), 016101 for the theory part.)”

Thank you for this input. We did not include such a plot, as one of the main objectives was the investigation of the limiting wavelength of impedance-matched absorption. Moreover, the underlying silicon nitride starts to significantly affect the reflection and absorption coefficients for very thin Au films. However, we think the reviewer has made a good point and we now included this plot in the supplementary information and mention it in the manuscript. In order to get meaningful values for the absorptivity and reflectivity, we chose to average the signal over the spectral range from 15 μ m to 20 μ m, in order to avoid the absorption peak of the supporting silicon nitride substrate.

The theory recommended by the reviewer (which in case of S. Zanotto et al. is also based on a transfer matrix) has been generally used to fit the supporting substrate and measured reflectivity and transmittance in fig.3,4 and generate the model in fig. 4 c.

“Then, they need to say how their work is different than those 2 references. I think it is because this work is in the IR and the 2 refs. are THz, but the authors need to make this conclusion, not me as reviewer. Especially since there is a lot of talk about THz in this paper.”

This is correct. We fabricated an absorber with a flat response over the entire near- and mid-IR range, limited to a wavelength of 2 μ m, whereas typically, impedance-matched absorbers are demonstrated in the far-IR (THz) regime.

Pham et al. present a graphene-based impedance-matched absorber for the far-IR (THz) regime compared to the mid-IR of our absorber. The fabrication and implementation of a graphene-based absorber on a large scale is more challenging than our method, which is based on standard physical vapor deposition of standard metals that allows for an easy fabrication on a large scale.

Brown et al. present an ITO-base impedance-matched absorber operating in the far-IR (THz) regime compared to the mid-IR of our absorber. In order to achieve the necessary sheet resistance, ITO thin films of the order of 50nm – 100nm are required, which is a lot thicker than the 2nm of Au required for our absorber.

We have revised the text in the introduction to clarify this point.

“A minor 3rd point is, what is the dip due to in the figure 3?”

Thank you for this point. Correct, this dip around 9 μm – 13 μm is a typical intrinsic absorption feature of the underlying silicon nitride. We revised the manuscript and now mention the source of this dip.

Reviewers' comments:

Reviewer #2 (Remarks to the Author):

I appreciate effort to consider my comments. However there are some changes which I believe necessary.

- The revised sentence "A comparison between seeded and unseeded membranes of this batch showed no significant influence on the electrical or optical properties by the oxidized seed layer." Is not clear. The authors should say as in their comment that they mean thickness. Something like the following? "A comparison between seeded and unseeded membranes of this batch showed no significant dependence of the electrical or optical properties on the oxidized seed layer thickness."

- I still have major concerns about the fact that they measure a sheet resistance at 0.5 nm gold thickness lower than that at 1 nm. How many measurements were they taken? I mean, how many sample runs? If the authors do not have enough statistics to support the fact that this unexpected "reverse" trend is real one possibility is to omit the data related to 0.5 nm. The observation may be related to some artefacts and they have to be sure it is not if they include it in the paper.

Reviewer #3 (Remarks to the Author):

The authors have revised the manuscript to address the concerns raised by my review and the other reviews.

I recommend publication.

Response Letter

Reviewer #2 Remarks and Answers

“I appreciate effort to consider my comments. However there are some changes which I believe necessary. The revised sentence “A comparison between seeded and unseeded membranes of this batch showed no significant influence on the electrical or optical properties by the oxidized seed layer.” Is not clear. The authors should say as in their comment that they mean thickness. Something like the following? “A comparison between seeded and unseeded membranes of this batch showed no significant dependence of the electrical or optical properties on the oxidized seed layer thickness.”

We thank the reviewer for the thoughtful suggestion and revised the sentence in the manuscript accordingly.

“I still have major concerns about the fact that they measure a sheet resistance at 0.5 nm gold thickness lower than that at 1 nm. How many measurements were they taken? I mean, how many sample runs? If the

authors do not have enough statistics to support the fact that this unexpected “reverse” trend is real one possibility is to omit the data related to 0.5 nm. The observation may be related to some artefacts and they have to be sure it is not if they include it in the paper.”

We are very grateful on this critical remark and want to thank the reviewer. In the course of this second revision, additional measurements of the samples below percolation and further statistical evaluation revealed that this data point (more precise at 0.35 nm) cannot be omitted. Indeed, the exact same reverse trend was observed in former studies of Au UTMFs (and other metals) grown on Cu modified Si [1,2]. Its origin lies in the increased surface roughness during the initial island growth. Before percolation occurs, those islands act as additional scattering centres that significantly increase the resistivity (see Supplement figure S3). Hence, the measured reverse trend is indeed a phenomenon of UTMFs below percolation. Our additional analysis is now presented and discussed in a new section in the Supplementary Information. And we also discuss the reverse effect in the manuscript.

Regarding the number of sample runs: we have fabricated Au UTMF with a thickness of 2 nm for more than five different oxidised Cu surfactant thicknesses. All of these independently fabricated 2 nm samples showed a highly reproducible behaviour. The analysis of UTMFs of different Au thickness is based on single deposition runs for each thickness. And our measurements represent multiple independent measurements of the samples of each Au thickness. We now provide all data details in the respective figure captions in compliance with Nature’s data reporting standard.

[1] D. V. Gruznev, D. A. Olyanich, D. N. Chubenko, D. A. Tsukanov, E. A. Borisenko, L. V. Bondarenko, M. V. Ivanchenko, A. V. Zotov, and A. A. Saranin, Growth of Au thin film on Cu-modified Si (1-1-1) surface, Surface Science 603, 3400 (2009).

[2] Z. Korczak and T. Kwapinsk, Electrical conductance at initial stage in epitaxial growth of Pb, Ag, Au and In on modified Si (1-1-1) surface, Surface Science 601, 3324 (2007).

Reviewer #3 Remarks and Answers

The authors have revised the manuscript to address the concerns raised by my review and the other reviews. I recommend publication.

We thank the reviewer for the final recommendation.

REVIEWERS' COMMENTS:

Reviewer #2 (Remarks to the Author):

Latest comments were properly addressed and paper can be published